# Preventive Effect of the Total Polyphenols from *Nymphaea candida* on Sepsis-Induced Acute Lung Injury in Mice via Gut Microbiota and NLRP3, TLR-4/NF-κB Pathway

**DOI:** 10.3390/ijms25084276

**Published:** 2024-04-12

**Authors:** Chenyang Li, Xinxin Qi, Lei Xu, Yuan Sun, Yan Chen, Yuhan Yao, Jun Zhao

**Affiliations:** 1School of Public Health, Xinjiang Medical University, Urumqi 830011, China; licy0609@126.com (C.L.); xjmuqxx@163.com (X.Q.); 2Xinjiang Key Laboratory for Uighur Medicine, Institute of Materia Medica of Xinjiang, Urumqi 830004, China; xulei19820218@126.com (L.X.); chenmaomao158@163.com (Y.C.); 17726840515@163.com (Y.Y.); 3School of Pharmacy, Xinjiang Medical University, Urumqi 830011, China; sym5658@163.com

**Keywords:** *Nymphaea candida*, polyphenols, acute lung injury, inflammatory response, NLRP3, TLR-4/NF-κB pathway, gut microbiota, short-chain fatty acids

## Abstract

This study aimed to investigate the preventive effects of the total polyphenols from *Nymphaea candida* (NCTP) on LPS-induced septic acute lung injury (ALI) in mice and its mechanisms. NCTP could significantly ameliorate LPS-induced lung tissue pathological injury in mice as well as lung wet/dry ratio and MPO activities (*p* < 0.05). NCTP could significantly decrease the blood leukocyte, neutrophil, monocyte, basophil, and eosinophil amounts and LPS contents in ALI mice compared with the model group (*p* < 0.05), improving lymphocyte amounts (*p* < 0.05). Moreover, compared with the model group, NCTP could decrease lung tissue TNF-α, IL-6, and IL-1β levels (*p* < 0.05) and downregulate the protein expression of TLR4, MyD88, TRAF6, IKKβ, IκB-α, p-IκB-α, NF-κB p65, p-NF-κB p65, NLRP3, ASC, and Caspase1 in lung tissues (*p* < 0.05). Furthermore, NCTP could inhibit ileum histopathological injuries, restoring the ileum tight junctions by increasing the expression of ZO-1 and occludin. Simultaneously, NCTP could reverse the gut microbiota disorder, restore the diversity of gut microbiota, increase the relative abundance of Clostridiales and Lachnospiraceae, and enhance the content of SCFAs (acetic acid, propionic acid, and butyric acid) in feces. These results suggested that NCTP has preventive effects on septic ALI, and its mechanism is related to the regulation of gut microbiota, SCFA metabolism, and the TLR-4/NF-κB and NLRP3 pathways.

## 1. Introduction

Acute lung injury (ALI) is a life-threatening disease, causing high morbidity and mortality worldwide, which is a clinical syndrome caused by various direct or indirect causative factors, including pathogenic microbial infections, and mainly manifested by acute progressive respiratory distress and intractable hypoxemia [1]. Acute inflammation of the lung interstitium in ALI can produce neutrophil infiltration and pro-inflammatory cell aggregation and, ultimately, impede the lung gas exchange. It is reported that the mortality rate of acute respiratory distress syndrome (ARDS) can be up to 40% [2,3]. For example, novel coronavirus (SARS-CoV-2) infection can trigger a cytokine storm, which, in turn, leads to severe ALI, ARDS, and even multi-organ failure until death [4,5]. Moreover, sepsis is a leading cause of death in intensive care unit (ICU) patients [6]. The progress of sepsis is rapid and causes damage to multiple organs, of which lung injury and failure are the earliest and most frequent occurrences. In recent years, Chinese medicines have played a greater role in the prevention and therapy of ALI induced by various causes, especially in the treatment of ALI/ARDS caused by SARS-CoV-2 infection, such as Qingfei Paidu Tang, Ma Xing Shi Gan Tang, and total flavonoids from Sanyeqing and baicalein [1,7,8,9].

It is reported that there is a close interaction between gastrointestinal health and the severity of sepsis-induced lung disease. The lungs and intestines can influence each other through the flora and the immune system to achieve a bidirectional regulatory effect [10]. The intestinal epithelium is a natural physical barrier that prevents the transfer of harmful intestinal bacteria and their metabolic byproducts, including lipopolysaccharide (LPS), into the bloodstream [11]. LPS-mediated pathological responses have been a hot spot; exogenous LPS can enter the portal circulation, causing an imbalance in the intestinal flora in vivo. Imbalanced and disturbed flora causes the exudation of endogenous LPS, which in turn destroys the intestinal mucosa, increases intestinal permeability, and permits the bioproducts of intestinal microbiota to be released in the intestinal tract, which activates the systemic immune system, where a large amount of LPS enters the lungs along with blood and exacerbates the immune damage to the lungs [12]. Animal models of sepsis usually use infection to elicit a systemic inflammatory response, such as intraperitoneal injection of LPS, to mimic the mechanisms of sepsis onset and progression in humans [5]. Serum-increased LPS binds to the toll-like receptor 4 (TLR4) complex, which has been shown to be the main signaling receptor for LPS-mediated injury in the body. NF-κB is an important signaling molecule in the signaling pathway downstream of TLR4 [13] and is also a key hub in the regulation of inflammatory pathways, which can increase the inflammatory response and activate inflammatory cascade reactions in ALI [14].

*Nymphaea candida* flower is a traditional Chinese medicine, with efficacy in moisturizing the lungs and relieving coughs, clear heat, and strengthening the heart, mainly distributed in the Bosten Lake, Yili, and Altay regions in Xinjiang, China [15,16]. *N. candida* is also one of the main medicinal materials in the Uyghur medicine compound “Zukamu granules”, and this compound preparation has played a significant role in the treatment of COVID-19 [17,18]. Zukamu granules could significantly reduce the apoptosis rate of alveolar macrophages, enhance phagocytosis of alveolar macrophages, and inhibit high expression of the TLR4/MyD88/NF-κB signaling pathway [19]. *N. candida* has a variety of biological activities, such as antibacterial, anti-inflammatory, hypotensive, hepatoprotective, and neuroprotective effects, and polyphenols are its main characteristic compounds [16,20,21]. Total polyphenols from *N. candida* (NCTP) have been previously prepared and showed better anti-inflammatory, analgesic, antipyretic, and cough activities [22]. However, whether it has the effect of preventing pneumonia has not been reported in the literature. Therefore, this study intends to investigate the preventive effects of NCTP on LPS-induced septic acute lung injury in mice via the gut microbiota, metabolism of SCFAs, and NLRP3 and TLR-4/NF-κB pathways. The result may provide a new way of thinking for the prevention of ALI.

## 2. Results

### 2.1. Compound Analysis of NCTP

Chemical compositions of NCTP were confirmed by UHPLC-HRMS based on their retention time (Rt), accurate molecular mass (mass error of less than 5 ppm), and major MS/MS fragment ions. As shown in Figure 1 and Table 1, a total of 14 compounds were identified and deduced, including 10 hydrolyzable tannins and 4 flavonoid glycoside compounds. Some compounds were identified compared with reference substances, such as isostrictiniin, ellagic acid, and nicotiflorin [22].

### 2.2. Protective Effect of NCTP on Lung Tissue in ALI Mice

In the experiment, the control group had a healthy fur status, good spirit, and normal defecation. After modeling, a large number of secretions appeared in the corners of the eyes in the model group, meaning that mice were unable to open their eyes, followed by messy fur, laziness, slowness, and poorly formed feces. The mice in the NCTP (50 mg/kg) group had a small amount of secretions in their eyes, which resulted in their eyes being slightly open, with messy fur and unshaped feces, while the mice in the NCTP (100, 200 mg/kg) group had occasional secretions in the eyes, and their fur and other appearances were close to the control group. The DEX (3 mg/kg) had occasional secretions in their eyes, messy fur, and symptoms of slow reaction. All groups showed weight loss after modeling, but there were no differences between all groups in terms of weight (Table 2). There were no deaths in any groups of mice. As a major feature of ALI, pulmonary oedema of lung tissue was reflected by the lung wet-to-dry (W/D) ratio. Compared with the control group, the lung W/D ratio in the model group was significantly increased after LPS stimulation (*p* < 0.01). In contrast, NCTP (100, 200 mg/kg) and DEX (3 mg/kg) could decrease the lung W/D ratio in LPS-induced ALI mice (*p* < 0.05 or *p* < 0.01) (Table 2).

We explored lung injury in each group, as shown in Figure 2. The lung tissues in the control group represented normal structure without histopathologic changes. Compared to the control group, the lung tissues in the model group exhibited severe injury as follows: alveoli, alveolar tubes, and bronchi were distorted; the connective tissue showed hyperplasia; a large amount of inflammatory cell infiltration was observed in the pulmonary interstitium. Compared with the model group, the lung tissues in the NCTP (50 mg/kg) group showed hyperplasia in a small amount of the connective tissue, inflammatory cell infiltration, and thickening of the alveolar wall; the lung tissues in the NCTP (100 mg/kg) group showed significant improvement, with a small amount of connective tissue hyperplasia and occasional mild inflammatory cell infiltration; the lung tissues in the NCTP (200 mg/kg) and DEX (3 mg/kg) groups did not show significant inflammatory infiltration and connective tissue hyperplasia. Non-parametric tests of pathological grading showed that the mean rank value of the model group was significantly higher compared with the control group (*p* < 0.01). Compared with the model group, the mean rank value of NCTP (100, 200 mg/kg) decreased significantly (*p* < 0.05, Table 3). The results showed that NCTP could improve pathologic changes in acute inflammation attenuating lung tissue injury.

As an indicator of inflammatory cell extravasation, the lung tissue MPO activities in the model group mice were significantly upregulated compared with the control group (*p* < 0.01). However, NCTP (100, 200 mg/kg) and DEX (3 mg/kg) could markedly reverse this change induced by LPS (*p* < 0.05 or *p* < 0.01). The results indicated that NCTP could inhibit the LPS-induced recruitment of inflammatory cells (Figure 3 and Appendix A).

### 2.3. Effect of NCTP on BALF BCA and LDH Activities in ALI Mice

As shown in Figure 4 and Appendix A, the BALF BCA and LDH activities were significantly enhanced in LPS-induced septic ALI mice compared to the control group (*p* < 0.05 or *p* < 0.01). Nevertheless, treatment of NCTP (200 mg/kg) could reverse LPS-induced increases in BCA (*p* < 0.05). NCTP (50, 100 and 200 mg/kg) and DEX (3 mg/kg) could markedly decrease the LDH activity in ALI mice (*p* < 0.01).

### 2.4. Effect of NCTP on Inflammatory Cells and Pro-Inflammatory Cytokines in ALI Mice

The white blood cell (WBC) count is a well-known basic parameter to reflect the host immune response, which would be increased under chronic or acute inflammation conditions [23]. In this study, there was a significant increase in the WBC count in the model group when compared to the control group (*p* < 0.01). Moreover, compared with the model group, treatments with NCTP (100, 200 mg/kg) displayed a significant decrease in the WBC count (*p* < 0.01). From the results of the routine blood test, compared with the control group, the levels of neutrophil count (NEU), monocyte count (MON), basophil count (BAS), and eosinophil count (EOS) were significantly increased in the model group (*p* < 0.05 or *p* < 0.01). NCTP (100, 200 mg/kg) improved the trend of the above indicators after intervention (*p* < 0.05 or *p* < 0.01). The level of lymphocyte count (LYM) decreased in the model group (*p* < 0.05), and the NCTP (200 mg/kg) could improve the decreasing trend (*p* < 0.05) (Figure 5A–F and Appendix A).

Next, we explored the plasma LPS content. The plasma LPS level in the model group was significantly increased compared to the control group (*p* < 0.01). The NCTP (50, 100 and 200 mg/kg) and DEX (3 mg/kg) groups could significantly decrease the plasma LPS level in ALI mice compared to the model group (*p* < 0.05 or *p* < 0.01) and showed a dose-dependent effect (Figure 6A and Appendix A). The lung tissue inflammatory levels of cytokines TNF-α, IL-6, and IL-1β 16 h post-LPS-treatment are shown in Figure 6B–D. The results showed that LPS induced significant increases in TNF-α, IL-6, and IL-1β production (*p* < 0.01), while the NCTP (50, 100 and 200 mg/kg) and DEX (3 mg/kg) groups reduced TNF-α, IL-6, and IL-1β levels in lung tissue (*p* < 0.01).

### 2.5. Effect of NCTP on NF-κB and NLRP3 Signaling Pathway Activation

Intestinal microflora dysfunction may induce G-bacteria to produce a large amount of LPS that potentially enters the circulation through intestinal leakage, upregulating the lung immune response and activating the TLR4 signaling pathway [12]. Therefore, we next explored how the gut flora regulated immunity in animals with LPS-induced ALI by regulating the TLR4/NF-κB signaling pathway [24].

In this study, the expressions of the TLR4/NF-κB pathways in lung tissues were detected by Western blotting. As shown in Figure 7 and Appendix A and Appendix A, the levels of TLR4, MyD88, TRAF6, and IKKβ proteins showed obvious increases in the model group (*p* < 0.01), which were attenuated after treatment of NCTP (200 mg/kg) (*p* < 0.05 or *p* < 0.01). In addition, LPS stimulation apparently induced the phosphorylation of NF-κB p65 and IκB-α (*p* < 0.01), whereas NCTP (200 mg/kg) (*p* < 0.05 or *p* < 0.01) administration reversed this result.

As another important signal pathway for modulating the inflammatory process, the NLRP3 signaling pathway has been widely studied in recent years. Therefore, to further investigate the anti-inflammatory mechanism of NCTP in LPS-induced ALI, we also measured NLRP3 signaling pathway activation by NLRP3, ASC, and Caspase1 proteins 16 h after LPS treatment.

The results showed that NCTP could inhibit LPS induced by these three major proteins in the NLRP3 signaling pathway in a dose-dependent manner in a mouse model of ALI. As described in Figure 8 and Appendix A and Appendix A, compared with the control group, the levels of NLRP3, ASC, and Caspase1 were significantly increased in the model group (*p* < 0.01), and the levels of NLRP3, ASC, and Caspase1 were strongly significantly inhibited by NCTP (200 mg/kg) (*p* < 0.05 or *p* < 0.01).

### 2.6. Protective Effect of NCTP on the Intestinal Mucosa

LPS administration altered the gut flora in mice, which resulted in slight swelling of the ileum, trifling edema, gut villus separation, and gut mucosal damage [25]. These changes were demonstrated in the model group but not in the control group. In contrast, the NCTP (200 mg/kg) group intervention ameliorated the intestinal injury caused by LPS administration when compared to the model group. Images of the pathological samples are shown in Figure 9A. The structures of crypts in the control group mice were obvious, showing a large number of goblet cells with mucin secreted in each crypt, and the colonic epithelium was covered with a thick mucus layer, which was continuous and intact (Figure 9B). The mice in the model group had a severe loss of mature goblet cells, a significant reduction in the number of goblet cells per crypt, and a destroyed mucus layer. Compared with the model group, the LPS-induced massive depletion of goblet cells and mucin was significantly ameliorated in the NCTP (200 mg/kg) groups.

As shown in the ileum tight junction structural morphology observed using an electron microscope (Figure 9C), regularly aligned microvilli, complete tight junctions, and intact organelles were observed in the intestinal epithelium in the control group. In the model group, the number of microvilli was reduced, lodging and shedding occurred, and their length and arrangement were irregular. At the same time, tight junctions were discontinuous, mitochondria were swollen, and vacuoles appeared between epithelial cells. In the NCTP (200 mg/kg) groups, the microvilli in the intestinal epithelium were regular, and the tight junctions tended to be intact with the mice in the model group. The LPS-binding proteins (LBPs) in intestinal tissue in the model group were highly expressed compared with the control group (*p* < 0.01). With NCTP (50,100 and 200 mg/kg) intervention, LBPs were decreased in comparison with the model group (*p* < 0.05 or *p* < 0.01) (Figure 9D and Appendix A).

NCTP restored the ileum induced by LPS. As shown in Figure 10, Appendix A and Appendix A and Appendix A, ZO-1 and occludin protein expressions were lower in the model group compared to the control group (*p* < 0.01). NCTP (50, 100, and 200 mg/kg) treatments had a greater expression of ZO-1 and occludin compared with the model group (*p* < 0.01).

### 2.7. NCTP Reversed the Dysbiosis of Microbiota Composition and the Reduction in SCFA Secretion in ALI Mice

The present results showed that the alpha diversity of the gut microbiome was significantly lower in the model group compared to the control group (*p* < 0.01) [26]. By contrast, the alpha diversity of the NCTP (200 mg/kg) group was significantly higher than the model group (*p* < 0.01) (Figure 11A–D and Appendix A). The beta diversity of gut microbiota was analyzed to compare the similarity of different samples in species diversity [27] (Figure 11E,F). There was a significate difference in beta diversity among the groups. The control group and NCTP (200 mg/kg) group were clearly separated with the model groups, suggesting that the microbial community structure was different between the model group and control group. To clarify which bacteria were responsible for the differences in the gut microbiota structure among the groups, we explored the detailed differences in the relative abundance of bacterial communities at the phylum level.

At the phylum level, the intestinal flora of the organism is dominated by Bacteroidetes, Firmicutes, Proteobacteria, and Actinobacteria, which are beneficial bacteria, except Proteobacteria [28]. The major trend in the microbial community reaction to LPS-induced ALI was the loss of Firmicutes and the bloom of Proteobacteria [29]. Proteobacteria, Gram-negative aerobic bacteria, were considered a sign of gut microbiota dysbiosis and an important source of LPS, which could induce inflammation and be associated with many chronic diseases [30]. The relative abundance of Firmicutes (control, 0.34%; model, 0.18%) was decreased (*p* < 0.01), whereas the relative abundance of Proteobacteria (control, 0.08%; model 0.32%) was increased (*p* < 0.01) compared to the control group (*p* < 0.01). Moreover, we explored the microbiota relative abundance in the NCTP (200 mg/kg) groups. The abundance of Firmicutes (NCTP, 0.48%; model, 0.18%) increased significantly (*p* < 0.01), whereas Proteobacteria (NCTP, 0.13%; model, 0.32%) decreased in the NCTP (200 mg/kg) group compared with the model group (*p* < 0.01). The relative abundance of Bacteroidetes has no variances in all groups. As shown in the Venn diagram, the number of characteristic bacteria in the model group was decreased compared with the control group; the number of characteristic bacteria in the NCTP (200 mg/kg) group was restored compared with the model group. Furthermore, the differential analysis of LDA effect size (LEfSe) showed that Clostridiales and Lachnospiraceae were the representative bacterial order and family present in the NCTP (200 mg/kg) group (Figure 12 and Appendix A). Lachnospiraceae belonged to Clostridiales in the order, which was the predominant species in NCTP.

Compared with the control group, the fecal acetic acid, propionic acid, and butyric acid contents in the model group were obviously reduced (*p* < 0.01). Fecal acetic acid, propionic acid, and butyric acid levels were higher in NCTP (200 mg/kg) group mice compared with the model group [31], respectively (*p* < 0.05 or *p* < 0.01). Differences in other SCFAs were not significant (Figure 13 and Appendix A).

In order to further investigate the association between mice intestinal flora and fecal SCFAs at the order level, the present study was based on the Spearman rank correlation test. We examined the correlation between the relative abundance of Clostridiales and Lachnospiraceae on the concentrations of three SCFAs (acetic acid, propionic acid, and butyric acid) (Figure 14). Spearman’s correlation coefficients of 0.5 ≤ r < 0.8 were considered as moderate correlations, and r ≥ 0.8 was considered as high correlations. Compared with the model group at the order level, acetic acid, propionic acid, and butyric acids were associated with Clostridiales, showing high positive correlation, as well as Lachnospiraceae at the family level.

## 3. Discussion

The pathogenesis of ALI involves several pathological processes, including immune cell overactivation, cytokine storm, and oxidative stress [32]. Intraperitoneal injection of LPS simulates the disease model of ALI caused by sepsis in intensive care units. ALI is a common complication of sepsis, and the most serious complication of sepsis is infectious shock, which has high morbidity and mortality. So, this experimental model has certain clinical significance [33,34]. As a severe pulmonary disease, ALI is closely related to the inflammatory/anti-inflammatory signaling pathways as well as the dysregulation of intestinal flora [35]. It has been reported that endothelial cell and alveolar epithelial injury, inflammatory cell influx into the alveolus, lung oedema, and increased inflammatory cytokines are related to the pathogenesis of ALI. Oedema is a very typical symptom of ALI, and we used the lung W/D ratio to quantify the direct effects of ALI. In this study, we found that NCTP could significantly reduce the lung coefficients of mice, attenuate pulmonary edema [36,37], and reduce the inflammatory cell infiltration of lung tissues and the pathological damage of lung tissues. We also found that NCTP can reduce the IL-1β, IL-6, and TNF-α expression levels in lung tissues and reduce the level of LPS in plasma [38], which suggests that NCTP can effectively ameliorate LPS-induced ALI.

Exogenous LPS enters the portal circulation, causing an imbalance in intestinal flora in vivo. Intestinal microflora disorders have been shown to induce G-bacteria to produce a large amount of LPS, which enters the circulation through gut leakage, increasing the immune response and activating the TLR4 signaling pathway in the lungs [39]. TLR4 has been shown to be the main signaling receptor for LPS-mediated injury in the body, where it has identified pathogenic bacteria that have translocated into the blood circulation, initiated downstream signal transduction pathways, activated the NF-κB pathway through the myeloid differentiation protein 88 (MyD88)-independent pathway and MyD88-dependent pathway, and upregulated the expression of various inflammatory factors [3]. In this study, NCTP suppressed the expression of TLR4, MyD88, TRAF6, IKKβ, p-IκBα, and p-NF-κB p65, which was upregulated by LPS [40]. These findings indicated that NCTP inhibited the activation of the TLR4/MyD88/NF-κB pathway in LPS-treated lung tissue, thereby suppressing NF-κB activation [41].

The production and secretion of the pro-inflammatory cytokine IL-1β is required not only by the NF-κB signaling pathway but also by the activation of the NLRP3 inflammasome. The NLRP3 inflammasome consists of an NLR family member, ASC, and Caspase-1 [42,43]. It can be activated by many factors, including bacteria and viruses, and evidence has suggested that the NLRP3 inflammasome is activated during ALI-induced LPS [44,45]. In the present study, we found that NCTP inhibited the proteins expression of NLRP3, ASC, and Caspase-1 in LPS-induced ALI mice. These results suggested that the protective effects of NCTP may derive from its direct suppression of NLRP3 inflammasome activation [46].

The gut houses the largest collection of bacteria and endotoxins in the human body [47]. Substantial research has supported that the gut microbiota is closely related to the development and progression of ALI by transmitting gut–lung axis activation. The intestinal flora plays an important role in maintaining the normal immune function of the body, reducing bacterial translocation and improving lung injury [48]. Compared to the control group, the model group with an imbalance in the gut flora showed more obvious gut damage and decreased expression of ZO-1 and occludin [29,49]. In contrast, NCTP was verified to reconstruct the gut flora, upregulate ZO-1 and occludin expression, and reduce gut damage. Thus, the imbalance in the gut flora exacerbated bacterial translocation and stimulated inflammatory cascades. However, the abundance of intestinal flora in the NCTP group was increased compared with the model group, and the structure was relatively stable after NCTP intervention, indicating that NCTP improved the imbalance in the gut flora induced by ALI administration and regulated the abundance and diversity of gut flora.

Firmicutes is a major component of the intestinal flora and the main group of bacteria affected by NCTP. A variety of flora under Firmicutes are probiotics, which are involved in the fermentation of polysaccharide components, the production of short-chain fatty acids, lactic acid and other components, and the maintenance of intestinal flora homeostasis. SCFAs are the most important metabolites of the intestinal flora, which are absorbed through the intestines and enter the circulatory system to play a role in regulating immunity [50]. Lachnospiraceae is the dominant SCFA-producing species in the Firmicutes. Zhang [51] speculated that gut dysbiosis is a major factor, especially a decrease in the abundance of butyrate-producing bacteria such as Lachnospiraceae. Members of Lachnospiraceae are able to utilize lactate and acetate to produce butyrate via the butyryl-CoA or acetate CoA transferase pathways or the butyrate kinase pathway. Moreover, supplementation with butyrate-producing bacteria may be beneficial for alleviating gut dysbiosis. The experimental results showed that NCTP significantly increased the acetic acid, propionic acid, and butyric acids levels. NCTP inhibited the relative abundance of harmful bacteria by enriching the SCFAs in beneficial bacteria through the correlation analysis between SCFAs and intestinal flora.

In summary, NCTP could alleviate sepsis-induced acute lung injury by regulation of the NLRP3 and TLR-4/NF-κB pathways and the gut microbiota. Polyphenols are the main active ingredients of NCTP, such as isostrictiniin, ellagic acid, and nicotiflorin. Isostrictiniin has better anti-inflammatory and antioxidant activities [16]; ellagic acid has better anti-inflammatory, antioxidant, and immunomodulatory properties [52,53]; nicotiflorin possesses antioxidant, antibacterial, anti-inflammatory, and analgesic activities [54]. However, the preventive effects of these components on ALI need further study to reveal the pharmacodynamic material basis of NCTP.

## 4. Materials and Methods

### 4.1. Chemicals and Reagents

LPSs (lipopolysaccharides from *Escherichia coli* O55: B5, L2880-100 mg) were purchased from Shanghai Aladdin Biochemical Technology Co., Ltd. (Shanghai, China). Dexamethasone Acetate tablets were purchased from Chongqing Kerui Pharmaceutical Group Co., Ltd. (Chongqing, China). PBS (2246306-500 mL) was obtained from Shanghai Darthel Biotechnology Co., Ltd. (Shanghai, China). Further, 4% paraformaldehyde fixative solution (BL539A) was purchased from Beijing Labgic Technology Co., Ltd. (Beijing, China). Hematoxylin and eosin staining solution (BA4097-500 mL) was obtained from BaSO Biotech Inc. (Zhuhai, China). RIPA buffer (R0010) and PMSF (P0100) were purchased from Beijing Solarbio Science & Technology Co., Ltd. (Beijing, China). Phosphatase Inhibitor Mix (PR20015) was purchased from Wuhan Sanying Inc. (Wuhan, China). BCA Protein Concentration Assay Kit (PC0020) was purchased from Beijing Solarbio Science & Technology Co., Ltd. (Beijing, China). LDH Kit (A020-1) was obtained from Nanjing Jiancheng Bioengineering Institute (Nanjing, China). ELISA kits for TNF-α (JL10484), IL-6 (JL20268), IL-1β (JL18442), LPS (JL18442), LBP (JL29644), and MPO (JL10367) were purchased from Shanghai Jianglai industrial Limited By Share Ltd. (Shanghai, China). NLRP3 antibody (ab270449), ASC antibody (ab309497), Caspase1 antibody (ab138483), MyD88 antibody (ab219413), TRAF6 antibody (ab33915), IKKβ antibody(ab124957), IκB antibody (ab32518), p-IκB antibody (ab133462), NF-κB p65 antibody (ab32536), p-NF-κB p65 antibody (ab76302), GAPDH antibody (ab181602), and occludin antibody (ab216327) were purchased from Abcam, Cambridge, UK. TLR4 antibody (#14358) was purchased from Cell Signaling Technology, Beverly, MA, USA. ZO-1 antibody (AF5145) was purchased from Affinity Biosciences (Changzhou, China).

### 4.2. Plant Materials and Preparation of NCTP

*Nymphaea candida* flower was purchased from Xinjiang Shengxiangcao Co. (Urumqi, China) and identified by Jiang He, a researcher at the Institute of Materia Medica of Xinjiang, and the voucher specimens (20210906N) were stored in the Institute of Materia Medica of Xinjiang. The extraction of NCTP was described in our previously published report [22]. This medicinal material was extracted with 70% ethanol by reflux three times for 1 h each time. We mixed the extract, concentrated under reduced pressure, and obtained the 70% ethanol extract. The extract was isolated by D101 resin, and 70% ethanol elutes were further purified using polyamide resins. The 70% ethanol elutes in enriched polyphenols were concentrated and dried to obtain NCTP.

### 4.3. Compound Analysis of NCTP

Compound analysis of NCTP was performed using a UHPLC-Q-Extractive mass spectrometry system consisting of an Ultimate 3000 UHPLC (Thermo Fisher Scientific, Bremen, Germany) combined with a Q Exactive Plus (Thermo Fisher Scientific, Germany). The chromatographic separation for NCTP was accomplished on an Agilent XDB C18 column (4.6 mm × 250 mm, 5 μm) (Agilent Technologies Inc., Santa Clara, CA, USA). The UHPLC flow rate was 1.0 mL/min, column temperature was 35 °C, and detection wavelength was 266 nm. The optimized mobile phases contained solvent A and solvent B, which were acetonitrile and 0.1% formic acid in water. The optimized mobile-phase gradient elution was as follows: 0~35 min, 5–15% A; 35–65 min, 15–18% A; 65~70 min, 18–20% A; 70–75 min, 20–5% A; 75–80 min, 5–5% A.

The detection parameters for the HRMS conditions were as follows: ESI spray voltage −2.8 kV (positive ions: 3.2 kV); sheath gas (sheath gas) 40 arb; auxiliary gas (auxiliary gas) 10 arb; curtain gas (CUR) 35; ion source temperature (HEAT TEMP) 350 °C; capillary temperature 300 °C; focusing voltage (FP) −350 V, DP2 −10 V. The whole process used nitrogen. Sample analysis was in DDA mode: Q-Orbitrap acquisition range was *m*/*z* 100–1500; fragment ion scanning range was set to *m*/*z* 50–1500; collision gas (NCE): 20%, 40%, 60%; MS resolution was 70,000 FWHM (*m*/*z* 200); MS^2^ resolution was 17,600 FWHM (*m*/*z* 200). The UHPLC-Q-Orbitrap-HRMS system operation and data analysis were carried out with the Thermo Fisher Scientific Workstation software, which contains Data Acquisition (Xcalibur 4.0) software and Qualitative Analysis (Xcalibur 4.0) software.

### 4.4. Experimental Animals

Seven- to eight-week-old C57/BL6J male mice were purchased from Vitalriver (Beijing, China). The mice were housed in isolated cages under controlled environmental conditions at the Xinjiang Medical University (12 h light–dark cycle, 55 ± 5% humidity, 25 ± 1 °C), with free access to standard laboratory food and water. All animal studies were approved by the Experimental Animal Ethics Committee of Xinjiang Medical University (Approval No: IACUC-20220314-2) and complied with animal welfare regulations. The methods employed in this study were conducted in accordance with the approved guidelines.

### 4.5. Sepsis-Induced Acute Lung Injury (ALI) Model

After one week of adaptive feeding, mice were randomly divided into six groups, with 12 mice in each group: control group, model group, DEX (positive control, 3 mg·kg^−1^) group, and dose NCTP (50, 100, 200 mg·kg^−1^). On days 1–7, mice underwent intragastric administration of NCTP (50, 100, 200 mg·kg^−1^) and DEX (3 mg·kg^−1^), respectively, except for the control and model groups with the same amount of 0.5% CMC-Na solution. After 30 min of the last administration, LPS (10 mg·kg^−1^) solution was intraperitoneal injection to induce ALI [23,55], except for the control group. Sixteen hours later, mice were euthanized, and bronchoalveolar lavage fluid (BALF), blood, lungs, ileum tissues, and fecal samples were performed, respectively.

### 4.6. Pathologic Analysis and Immunohistochemical Analysis

Histologic examination. The collected right upper lobe was fixed overnight in 4% paraformaldehyde, dehydrated, paraffin-embedded, cut into 5 µm thick sections, and stained with hematoxylin and eosin (H&E). Images were acquired using a light microscope (LEICA DM 3000, Wiesler, Hesse, Germany) at 20× magnification. The extent of lung damage was assessed according to previous studies [23]. The degree of lung injury was graded from 0 (normal) to 3 (severe) as follows: 0, no injury; Ⅰ, mild injury; Ⅱ, moderate injury; Ⅲ, severe injury. Individual scores for each category and the sum of the scores of 6 mice in each category were calculated to determine the total lung injury score for histological assessment. The collected ileum tissue was fixed overnight in 4% paraformaldehyde, dehydrated, paraffin-embedded, cut into 5 µm thick sections, and stained with hematoxylin and eosin (H&E). AB-PAS staining. Mouse colon tissue was taken and fixed with Carnot’s fixative, followed by paraffin embedding, sectioning, dewaxing, and staining with AB-PAS.

Ultrastructural analysis. Ileum tissue samples were immediately fixed with 3% glutaraldehyde immediately and stored at a temperature of 4 °C for 4 h. The samples were dehydrated with a graded series of ethanol and propylene oxide and then embedded in an epoxy resin. Ultrathin sections were cut, stained with uranyl acetate and lead citrate, and examined with transmission electron microscopy (Hitachi HT7800, Tokyo, Japan). The ultrastructure of the ileum tight junctions was evaluated by capturing high-magnification images.

Immunofluorescence and immunohistochemical analysis. Sections cut from paraffin-embedded ileum tissues were deparaffinized, washed with xylene and graded series of ethanol, subsequently recovered by antigen retrieval buffer with 10 mM citrate buffer, blocked with 10% goat serum, and incubated with primary antibody against ZO-1 (1/200), and occludin (1/200) at 4 °C overnight. Image J software (v1.8.0.112) was used for quantification.

### 4.7. BALF Collection, Processing, and Determination of Protein and LDH Concentration

The total protein content and LDH in bronchoalveolar lavage fluid (BALF) were examined. At the end of the experiment, mice were euthanized by cervical dislocation, 500 µL PBS was instilled twice through the trachea for 30 s, and BALF was collected. The collected BALF was used to determine the total protein content and LDH in the BALF supernatant using the BCA method and microplate spectrophotometer (Thermo Fisher Scientific, Multiskan GO, Waltham, MA, USA).

### 4.8. Measurement of Wet-to-Dry Ratio of the Lungs

Lung wet/dry weight ratio. The middle lobes of the right lung of mice were collected and rinsed with PBS. Excess surface water was removed by blotting with filter paper, and the lung tissue was then weighed using a sophisticated electronic balance to determine lung wet weight. Subsequently, the lung tissue was placed in a desiccator at a temperature of 80 °C and dried thoroughly for 72 h until a stable weight was obtained. The weight of the dried lung tissue was recorded as the lung dry weight. Finally, the lung wet/dry (W/D) weight ratio was calculated to assess the degree of pulmonary edema in mice.

### 4.9. Cell Count and Biochemical Assays

The white blood cells (WBCs), neutrophils (NEU), lymphocytes (LYM), monocytes (MON), eosinophils (EOS), and basophils (BAS) were detected by an Auto Hematology Analyzer (Mindray Healthcare International Inc. Mindray BC-5000, Shenzhen, China). The plasma LPS was determined by ELISA kits.

### 4.10. Determination of Lung Homogenate TNF-α, IL-1β, and IL-6

Lung homogenates were directly used to determine inflammatory mediators, including tumor necrosis factor α (TNF-α), interleukin-1β (IL-1β), and interleukin-6 (IL-6), by ELISA kits.

### 4.11. Analysis of Gut Microbiota

The cecum contents in different groups were collected, and total genomic DNA was extracted from each fecal sample with Stool DNA mini kits (Qiagen, Valencia, CA, USA). The purity and concentration of the extracted DNA were detected by gel electrophoresis; the DNA was stored at −80 °C for further analysis. The sample DNA was amplified by PCR amplification to enrich the V3-V4 region of 16S rDNA genes in bacteria. PCR products were quantified by a microplate reader (BioTek, FLx800, Winooski, VT, USA) and an Agilent Bioanalyzer 2100 system (Agilent Technologies Inc., Santa Clara, CA, USA). The whole genome sequencing was performed by double-end sequencing using a NovaSeq sequencer with NovaSeq 6000 SP Reagent Kit (500 cycles) (San Diego, CA, USA). The chao indices reflect the species richness, and the Shannon and Simpson indices reflect the species richness and uniformity in the sample. They were used to assess the alpha diversity among groups. The clustering characteristics of the gut microbiota were analyzed by beta diversity based on OTU abundances.

### 4.12. Fecal Short-Chain Fatty Acid (SCFA) Quantification Analysis

The concentrations of SCFAs were determined in the fecal supernatant using GC-MS (Thermo Trace 1300-Thermo ISQ 7000). Samples were homogenized for 1 min with 500 μL of water and 100 mg of glass beads, and then centrifuged at 4 °C for 10 min at 12,000 rpm. Then, 200 μL supernatant was extracted with 100 μL of 15% phosphoric acid and 20 μL of 375 μg/mL 4-methylvaleric acid solution as IS and 280 μL ether. Subsequently, the samples were centrifuged at 4 °C for 10 min at 12,000 rpm after vortexing for 1 min, and the supernatant was transferred into the vial prior to GC-MS analysis.

### 4.13. Western Blot

At the end of the experiment, 30 mg of the lower lobe of the right lung was weighed, and the total protein was extracted with RIPA lysis buffer. The protein was then quantified using the BCA method. Subsequently, 25 µg of protein was electrophoresed on an 8% denaturing polyacrylamide gel and transferred to a PVDF membrane. After blocking with the NcmBlot blocking buffer, the membrane was incubated overnight at 4 °C with anti-TLR4 (1:1000), MyD88 (1:1000), TRAF6 (1:5000), IKKβ (1:5000), IκB (1:2000), p-IκB (1:1000), NF-κB p65 (1:1000), p-NF-κBp 65 (1:1000), NLRP3 (1:1000), ASC (1:1000), Caspase-1 (1:1000), and GAPDH (1:1000). Then, it was incubated for another 1 h with horseradish peroxidase-conjugated rabbit anti-mouse IgG antibody (1:10,000). Finally, a chemiluminescence detection reagent and imaging system were used to visualize the membranes.

### 4.14. Statistical Analysis

Each experiment was conducted in triplicate, and the data are presented as mean ± standard deviation. Statistical comparisons between two groups were analyzed using the *t*-test, while differences among multiple groups were analyzed using one-way analysis of variance (ANOVA). GraphPad software version 9 (Prism, La Jolla, CA, USA) was utilized for statistical analysis. A *p*-value < 0.05 was considered statistically significant.

## 5. Conclusions

The experimental results suggest that NCTP could be considered as an anti-inflammatory agent for the treatment of ALI, and these findings can also confirm the traditional application of *N. candida* as well as its effects in classic formulas, including zukamu granules. This research could provide a reference for the development and application of the *N. candida* flower.

## Figures and Tables

**Figure 1 ijms-25-04276-f001:**
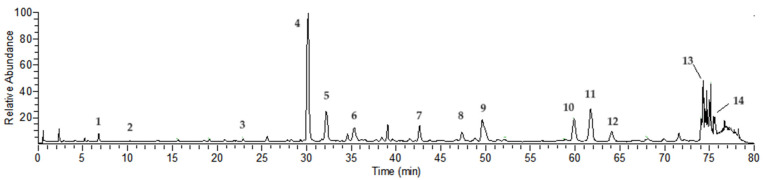
UHPLC-Q-Orbitrap-HRMS chromatogram of NCTP (ESI-negative mode). (The number in the figure caption means peak number).

**Figure 2 ijms-25-04276-f002:**
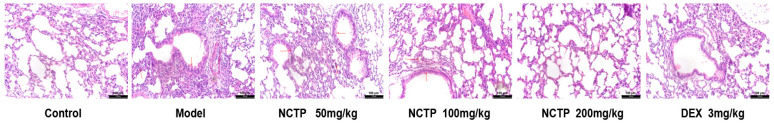
Representative images of H&E-stained lung sections in mice (*n* = 6) at 16 h after LPS administration, demonstrating alveolar structure. Scale bar, 100 µm.

**Figure 3 ijms-25-04276-f003:**
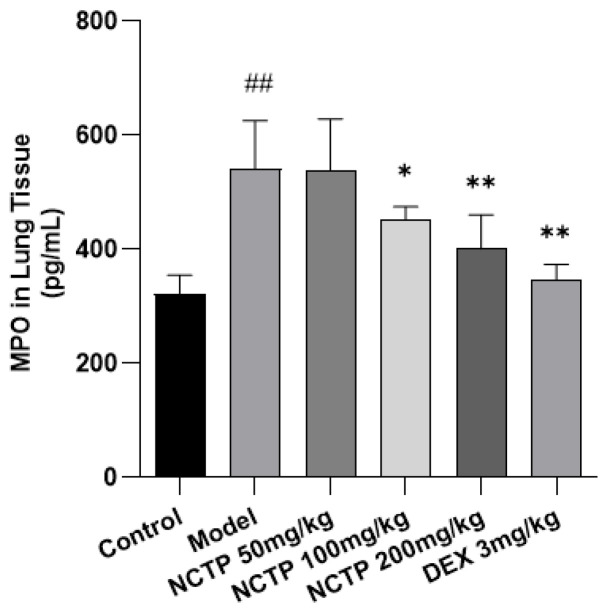
Effect of NCTP on lung tissues MPO levels in ALI mice. Data were presented as mean ± SD, *n* = 6. ^##^
*p* < 0.01, versus control group; * *p* < 0.05, ** *p* < 0.01 versus model group.

**Figure 4 ijms-25-04276-f004:**
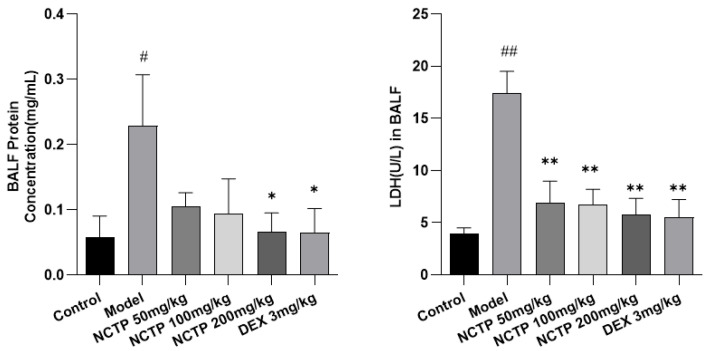
Effect of NCTP on BALF total protein; LDH contents in ALI mice. Data were presented as mean ± SD, *n* = 6. ^#^
*p* < 0.05, ^##^
*p* < 0.01, versus control group; * *p* < 0.05, ** *p* < 0.01 versus model group.

**Figure 5 ijms-25-04276-f005:**
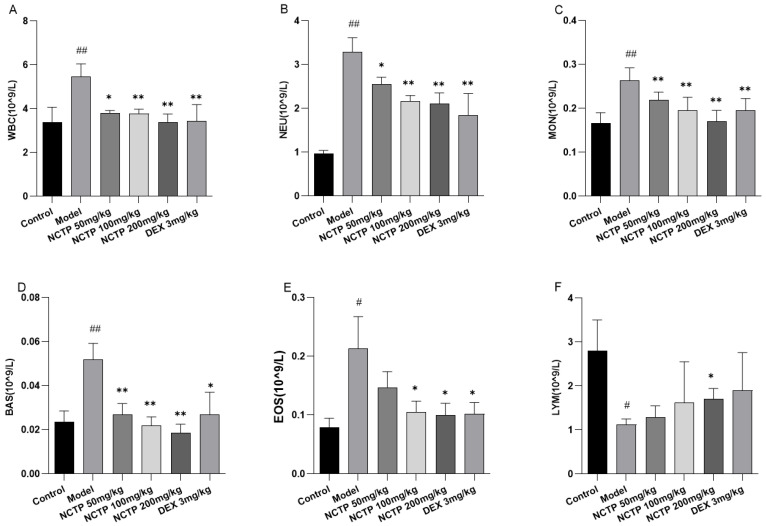
Effects of inflammatory factors in blood in ALI mice. (**A**) WBC count in blood in mice. (**B**) NEU count in blood in mice. (**C**) MON count in blood in mice. (**D**) BAS count in blood in mice. (**E**) EOS count in blood in mice. (**F**) LYM count in blood in mice. Data were presented as mean ± SD, *n* = 6. ^#^
*p* < 0.05, ^##^
*p* < 0.01, versus control group; * *p* < 0.05, ** *p* < 0.01 versus model group.

**Figure 6 ijms-25-04276-f006:**
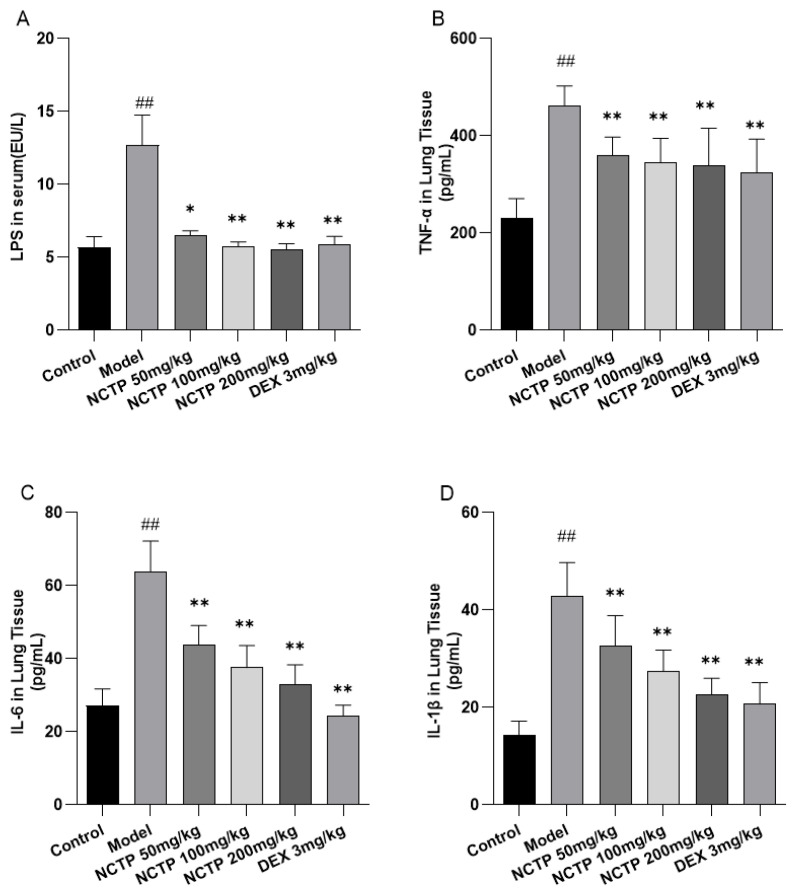
Effects of NCTP on inflammatory factors in ALI mice. (**A**) The levels of LPS in plasma of LPS-induced ALI mice. (**B**–**D**) The levels of TNF-α, IL-6, IL-1β in lung tissues of LPS-induced ALI mice. Data were presented as mean ± SD, *n* = 6. ^##^
*p* < 0.01, versus control group; * *p* < 0.05, ** *p* < 0.01, versus model group.

**Figure 7 ijms-25-04276-f007:**
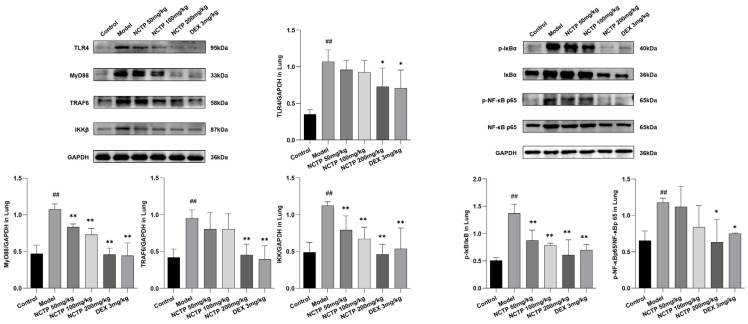
Effects of NCTP on proteins expression of lung tissues TLR-4/NF-κB in ALI mice. Data were presented as mean ± SD, *n* = 6. ^##^
*p* < 0.01, versus control group; * *p* < 0.05, ** *p* < 0.01 versus model group.

**Figure 8 ijms-25-04276-f008:**
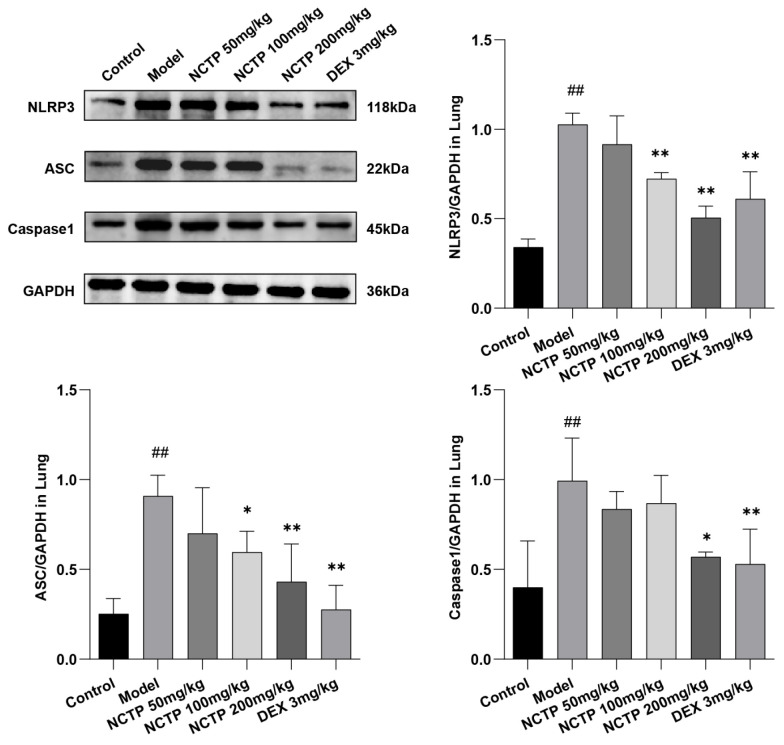
Effects of NCTP on proteins expression of lung tissues NLRP3 in ALI mice. Data were presented as mean ± SD, *n* = 6. ^##^
*p* < 0.01, versus control group; * *p* < 0.05, ** *p* < 0.01 versus model group.

**Figure 9 ijms-25-04276-f009:**
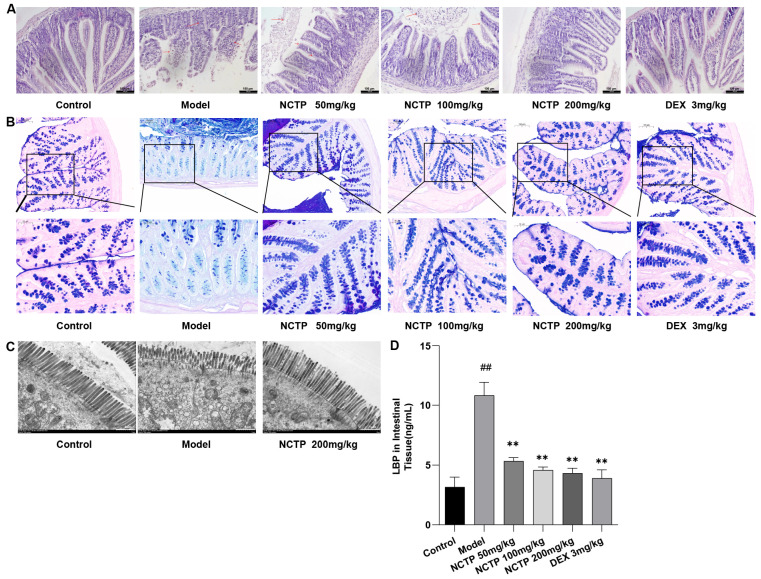
Effects of NCTP on the intestinal mucosa in ALI mice. (**A**) Effects of NCTP on histomorphology in ileum tissue of LPS-induced ALI mice. Scale bar, 100 µm. (**B**) Effect of NCTP in ileum tissue of LPS-induced ALI mice (AB-PAS stain) (at 10× magnification and 20× magnification). (**C**) Ultrastructural analysis in ileum tissue by transmission electron microscopy (at 1000× magnification). (**D**) Effect of NCTP on the levels of LBP in intestinal tissue. Data were presented as mean ± SD, *n* = 6. ^##^
*p* < 0.01, versus control group; ** *p* < 0.01 versus model group.

**Figure 10 ijms-25-04276-f010:**
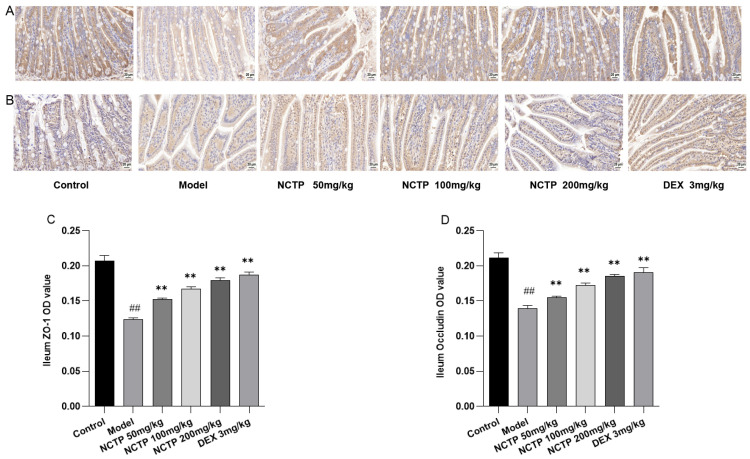
Effects of NCTP on ZO-1 and occludin protein expression in ALI mice. Immunohistochemical staining for ZO-1 (**A**,**C**) and occludin (**B**,**D**) in the ileum tissue (at 40× magnification). Data were presented as mean ± SD, *n* = 6. ^##^
*p* < 0.01, versus control group; ** *p* < 0.01 versus model group.

**Figure 11 ijms-25-04276-f011:**
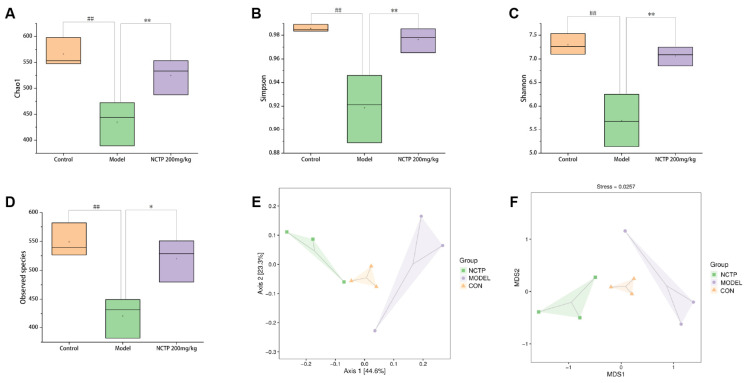
Effects of NCTP on the gut microbiota dysbiosis induced by ALI. The chao (**A**), Simpson (**B**), Shannon (**C**), and observed species (**D**) indices were used to assess the alpha diversity among groups. (**E**,**F**) The beta diversity among groups. Data were presented as mean ± SD, *n* = 6. ^##^
*p* < 0.01, versus control group; * *p* < 0.05, ** *p* < 0.01 versus model group.

**Figure 12 ijms-25-04276-f012:**
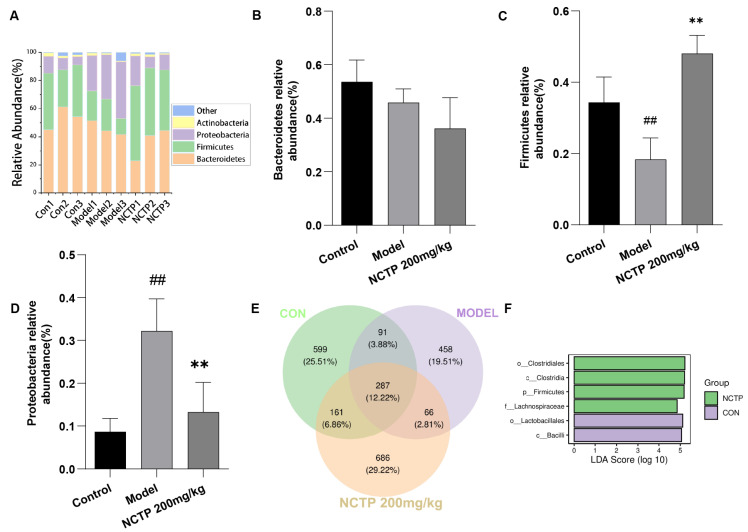
Effect of NCTP on phylum levels of gut microbiota. (**A**) The composition of gut microbiota of community bar plot analysis on phylum level. (**B**) The Bacteroidetes, (**C**) Firmicutes, (**D**) Proteobacteria on phylum level. (**E**) Venn diagram illustrated the overlap of OTUs in all tested mice. (**F**) Distribution histograms showing the results of LEfSe analysis. Data were presented as mean ± SD, *n* = 6. ^##^
*p* < 0.01, versus control group; ** *p* < 0.01 versus model group.

**Figure 13 ijms-25-04276-f013:**
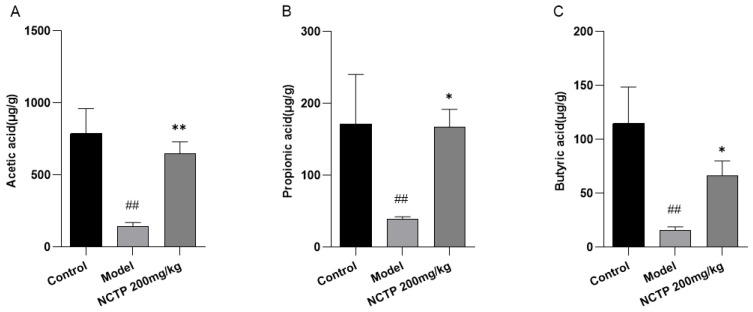
Effect of NCTP in the fecal content levels of short-chain fatty acids. (**A**) Acetic acid, (**B**) Propionic acid, (**C**) Butyric acid, (**D**) Isobutyric acid, (**E**) Valeric acid, (**F**) Isovaleric acid contents in fecal. Data were presented as mean ± SD, *n* = 6. ^#^
*p* < 0.05, ^##^
*p* < 0.01, versus control group; * *p* < 0.05, ** *p* < 0.01 versus model group.

**Figure 14 ijms-25-04276-f014:**
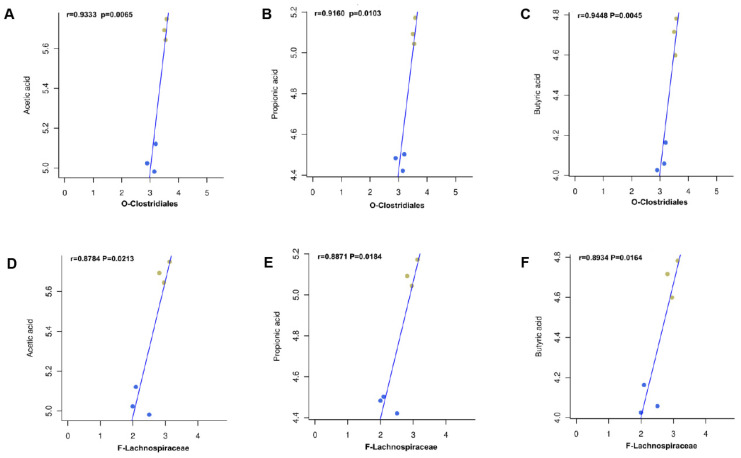
(**A**–**F**) Correlation analysis of the relative abundance of gut microbiota and fecal short-chain fatty acids. (The brown dots represent NCTP, 200 mg/kg, the blue dots represent model group).

**Table 1 ijms-25-04276-t001:** MS and MS/MS data of NCTP.

Peak No	Rt (min)	[M-H]^−^	Error	MS/MS *m*/*z* (% of Base Peak)	Formula	Compound Identity
1 ^a^	6.75	169.01385	4.17	125 (100)	C_7_H_6_O_5_	Gallic acid
2	10.29	483.07758	1.33	331 (20), 169 (100), 125 (80)	C_20_H_20_O_14_	Digalloyl-glucoside
3	21.87	635.08838	0.77	483 (40), 313 (20), 169 (100), 125 (80)	C_27_H_24_O_18_	Trigalloyl-glucoside
4 ^a^	30.19	633.07281	0.91	463 (20), 301 (100), 275 (20), 185 (10)	C_27_H_22_O_18_	Isostrictiniin
5 ^a^	32.15	197.04503	0.77	169 (100), 125 (50)	C_9_H_10_O_5_	Gallic acid ethyl ester
6	35.44	785.08282	−0.48	633 (10), 301 (100), 275 (50), 257 (10), 169 (40), 125 (30)	C_34_H_26_O_22_	Digalloyl-ellagitannoyl-glucoside
7	42.54	787.09906	0.26	635 (20), 617 (70), 465 (40), 313 (30), 169 (100), 125 (60)	C_34_H_28_O_22_	Tetragalloyl-glucoside
8	47.28	433.04095	1.85	301 (100), 271 (5), 243 (5)	C_19_H_13_O_12_	Quercetin-O-arabinoside
9 ^a^	49.69	300.99802	0.41	283 (20), 229 (10), 185 (20)	C_14_H_6_O_8_	Ellagic acid
10 ^a^	59.92	939.10931	−0.53	787 (20), 769 (100), 465 (10), 313 (10), 169 (90), 125 (80)	C_41_H_32_O_26_	Pentagalloyl-glucoside
11 ^a^	61.66	593.15015	0.08	285 (80), 255 (60), 227 (40), 151 (10)	C_27_H_30_O_15_	Nicotiflorin
12 ^a^	64.07	447.09293	1.65	285 (60), 255 (70), 227 (50), 151 (10)	C_21_H_20_O_11_	Kaempferol O-glucoside
13	74.66	447.09305	1.92	300 (100), 271 (70), 227 (50), 169 (100), 125 (20)	C_21_H_20_O_11_	Ellagitannoyl-rhamnoside
14 ^a^	75.47	285.03970	1.18	257 (10), 229 (10), 151 (10)	C_15_H_10_O_6_	Kaempferol

Rt, retention time; ^a^, compared with reference substance.

**Table 2 ijms-25-04276-t002:** Influence of NCTP on body weight in ALI mice.

Group	Pre-Modeling Weight (g)	Post-Modeling Weight (g)	W/D Ratio
Control	24.72 ± 0.58	22. 54 ± 0.51	3.96 ± 0.46
Model	23.38 ± 1.32	20.48 ± 1.29	5.62 ± 0.42 ^##^
NCTP, 50 mg/kg	23.08 ± 0.82	21.28 ± 0.66	4.37 ± 0.77
NCTP, 100 mg/kg	22.90 ± 0.69	20.76 ± 0.48	4.26 ± 0.66 *
NCTP, 200 mg/kg	23.01 ± 1.25	21.14 ± 1.10	4.38 ± 0.33 **
DEX, 3 mg/kg	23.04 ± 1.56	21.22 ± 1.10	4.12 ± 0.30 **

Data are presented as mean ± SD, *n* = 6. ^##^
*p* < 0.01 versus control group, * *p* < 0.05 and ** *p* < 0.01 versus model group.

**Table 3 ijms-25-04276-t003:** The pathological grading results of lung tissues in ALI mice.

Group	*n*	The Degree of Lung Injury				Rank Mean
		0	Ⅰ	Ⅱ	Ⅲ	
Control	6	6	0	0	0	9.50
Model	6	0	0	6	0	33.50 ^##^
NCTP, 50 mg/kg	6	0	6	0	0	24.50
NCTP, 100 mg/kg	6	3	3	0	0	17.00 *
NCTP, 200 mg/kg	6	4	2	0	0	14.50 **
DEX, 3 mg/kg	6	5	1	0	0	12.00 **

*n* = 6. ^##^
*p* < 0.01 versus control group, * *p* < 0.05 and ** *p* < 0.01 versus model group

## Data Availability

Data is contained within the article and Appendix A.

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
