# Peer review of "Preventive Effect of the Total Polyphenols from Nymphaea candida on Sepsis-Induced Acute Lung Injury in Mice via Gut Microbiota and NLRP3, TLR-4/NF-κB Pathway"

_ijms, 2024, doi:10.3390/ijms25084276_

Round 1

Reviewer 1 Report

Comments and Suggestions for Authors

Comments and questions:

Could you provide more context regarding the significance of Nymphaea candida and its total polyphenols in the context of septic acute lung injury? What led to the selection of this particular plant extract for the study?

How were the experiments conducted to assess the preventive effects of NCTP on septic acute lung injury in mice? Were there any specific dosages or treatment protocols used?

Can you elaborate on the mechanisms underlying the observed effects of NCTP on gut microbiota, SCFAs metabolism, and the Toll-like receptor 4/nuclear factor kappa-B (TLR-4/NF-κB) and NLRP3 pathways? Were these mechanisms hypothesized before the study, or were they discovered during the experimental process?

What are the potential implications of these findings for the development of novel preventive or therapeutic strategies for septic acute lung injury? Are there any limitations to the study that should be considered when interpreting the results?

It is not possible to see from the Figure 1 which compounds are identified?

Table 1: the names are technically not put correctly.

The future insight should be given in the conclusion part.

Reviewer 2 Report

Comments and Suggestions for Authors

The paper submitted by Li et al. investigates the in vivo effect of polyphenols from Nymphaea candida on sepsis induced by acute lung injury. Manuscript is clear, well written and the conclusions are supported by the results. However, some minor corrections are needed:

1. the introduction section could be completed with some references about ongoing clinical studies (if there are any).

2. fig 1 must be corrected; the spectra are not visible.

3. more specific results should be provided in the conclusion section. Which is the advancement brought by this study to the field?

4. some perspectives could also be given. 

Round 2

Reviewer 1 Report

Comments and Suggestions for Authors

The manuscript can be accepted.